# Drug-Loaded Polymeric Micelles Based on Smart Biocompatible Graft Copolymers with Potential Applications for the Treatment of Glaucoma

**DOI:** 10.3390/ijms23169382

**Published:** 2022-08-19

**Authors:** Manuela-Ramona (Blanaru) Ozturk, Marcel Popa, Delia Mihaela Rata, Anca Niculina Cadinoiu, Frederique Parfait, Christelle Delaite, Leonard Ionut Atanase, Carmen Solcan, Oana Maria Daraba

**Affiliations:** 1Faculty of Medicine, “Grigore T. Popa” University of Medicine and Pharmacy, 700115 Iasi, Romania; 2“Cristofor Simionescu” Faculty of Chemical Engineering and Environmental Protection, “Gheorghe Asachi” Technical University of Iasi, 700050 Iasi, Romania; 3Academy of Romanian Scientists, 050045 Bucharest, Romania; 4Faculty of Medical Dentistry, “Apollonia” University of Iasi, 700511 Iasi, Romania; 5LPIM, Université de Haute Alsace, 68100 Mulhouse, France; 6Public Health Department, Faculty of Veterinary Medicine, “Ion Ionescu de la Brad” University of Life Sciences, 700490 Iasi, Romania

**Keywords:** glaucoma, intraocular pressure, polymeric micelles, dorzolamide, indomethacin, cytotoxicity, haemolysis, in vivo tests

## Abstract

Glaucoma is the second leading cause of blindness in the world. Despite the fact that many treatments are currently available for eye diseases, the key issue that arises is the administration of drugs for long periods of time and the increased risk of inflammation, but also the high cost of eye surgery. Consequently, numerous daily administrations are required, which reduce patient compliance, and even in these conditions, the treatment of eye disease is too ineffective. Micellar polymers are core–shell nanoparticles formed by the self-assembly of block or graft copolymers in selective solvents. In the present study, polymeric micelles (PMs) were obtained by dialysis from smart biocompatible poly(ε-caprolactone)-poly(N-vinylcaprolactam-co-N-vinylpyrrolidone) [PCL-g-P(NVCL-co-NVP)] graft copolymers. Two copolymers with different molar masses were studied, and a good correlation was noted between the micellar sizes and the total degree of polymerisation (DPn) of the copolymers. The micelles formed by Cop A [PCL120-g-P(NVCL507-co-NVP128)], with the lowest total DPn, have a Z-average value of 39 nm, whereas the micellar sizes for Cop B [PCL120-g-P(NVCL1253-co-NVP139)] are around 47 nm. These PMs were further used for the encapsulation of two drugs with applications for the treatment of eye diseases. After the encapsulation of Dorzolamide, a slight increase in micellar sizes was noted, whereas the encapsulation of Indomethacin led to a decrease in these sizes. Using dynamic light scattering, it was proved that both free and drug-loaded PMs are stable for 30 days of storage at 4 °C. Moreover, in vitro biological tests demonstrated that the obtained PMs are both haemo- and cytocompatible and thus can be used for further in vivo tests. The designed micellar system proved its ability to release the encapsulated drugs in vitro, and the results obtained were validated by in vivo tests carried out on experimental animals, which proved its high effectiveness in reducing intraocular pressure.

## 1. Introduction

Glaucoma includes all of the pathological processes and ocular lesions linked to an abnormal increase in intraocular pressure (IOP), and it is generally an age-related disease that can lead to vision loss [1]. The goal of treatment is to permanently lower IOP. However, this treatment stabilises the disease but does not allow the regression of existing lesions. The first choice for the current treatment of glaucoma is the topical delivery of drugs, but these drugs have important drawbacks, such as poor bioavailability in the eye, which prevents them from reaching an effective drug concentration [2]. Consequently, numerous daily administrations are required in order to reach the therapeutic threshold, and even in these conditions, the treatment of eye disease is too ineffective [3]. The treatment of glaucoma is generally based on reducing the rate of aqueous humour secretion, and an important class of drugs is represented by carbonic anhydrase inhibitors (CAI). These drugs, classified into five groups (parasimpathomimetics such as pilocarpine, alpha2-agonists such as brimonidine, beta-blockers such as timolol, carbonic anhydrase inhibitors such as dorzolamide, and prostaglandin analogues such as latanoprost), can increase the rate of filtration of aqueous humour through the trabecular meshwork or uveoscleral pathways, leading to the reduced production of aqueous humour from ciliary processes [4].

Dorzolamide is usually administered three times a day for the treatment of glaucoma. It decreases the production of aqueous humour by crossing the cornea and reaching the ciliary body of the eye, leading to systemic effects on the carbonic anhydrase enzyme within the eye [5].

Indomethacin (IMC) is a nonsteroidal anti-inflammatory drug commonly used as a prescription medication to reduce fever, pain, stiffness, and swelling from inflammation, and it has also been used for the treatment of inflammation induced by glaucoma [6,7,8,9].

Although all of them reduce IOP, none are exempt from side effects, such as blurred vision, tachycardia, or arrhythmia.

To overcome these inconveniences and also the side effects generated by the administration of free drugs, different types of carriers, including hydrogels, nanoparticles, nanoemulsions, liposomes, and micelles, have been investigated in order to increase the ocular permeability of drugs, thus providing enhanced ocular drug bioavailability by increasing the corneal contact time [10,11,12,13,14,15].

The core–shell structure of polymeric micelles (PMs) is a valuable asset in improving the apparent solubility of drugs used for ophthalmic applications, as it can lead to controlled and sustained drug release since the dilution factor in the lachrymal fluid is less than in other administration routes [16]. In a recent review article, Durgun et al. highlighted the efficiency of in vitro, ex vivo, and in vivo studies concerning the efficiency of micellar systems for the treatment of glaucoma [17]. Nevertheless, from this review article, it appears that no studies exist concerning the loading of Dorzolamide in PMs. On the contrary, IMC has been loaded into different types of PMs.

IMC-loaded poly-(ethylene oxide)-poly(β-benzyl L-aspartate) (PEO−PBLA) micelles were prepared by dialysis [18]. Drug loading led to an increase in micellar size from 19 nm to around 25 nm. The drug release kinetics can be modulated by the pH. Another type of diblock copolymer, methoxy-poly(ethylene glycol)-b-poly(ε-caprolactone) (mPEG-PCL), was used by Kim et al. for the loading of IMC as well [19]. Based on the weight ratio between the drug and the copolymer, the drug loading efficiency (DLE) ranged between 16.33 and 42%. As previously reported, the micellar sizes increased from 120 nm to 145 and 165 nm for feed weight ratios of copolymer to IMC of 1:0.5 and 1:1. Finally, the drug release rate decreased with the increasing molecular weight of the copolymer.

Ouahab et al. used methoxy-poly(ethylene glycol)-b-poly(d, l-lactide) copolymer (mPEG-PDLLA) for the preparation of IMC-loaded solidified PMs (SPMs) and their in vivo investigation [20]. The pharmacokinetic parameters of the SPMs were found to be higher than commercial IMC.

An IMC–dextran amphiphilic conjugate was synthesised by Wand et al. [21], and the authors demonstrated the pH sensitivity of these micelles, having dextran as the corona and IMC as the core, due to the presence of disulphide bonds. These micelles were further used for the encapsulation of paclitaxel. Dextran stearate was used in another study for the preparation of IMC-loaded micelles by a dialysis method [22]. However, the size of these micelles was quite high, ranging between 313 and 600 nm. According to in vivo tests, it appears that these micelles are efficient in reducing inflammation in rats.

Very recently, a commercial copolymer, poly(vinyl caprolactam)-poly(vinyl acetate)-poly(ethylene glycol) graft copolymer, known under the name Soluplus^®^ from BASF GmbH (Munchen, Germany), was used for the preparation of PMs by a spray-drying process [23]. The size of these micelles was in the range of 100 to 500 nm, and the encapsulation efficiency was higher than 80%.

The aim of this study was to prepare and to characterise both Dorzolamide- and IMC-loaded smart PMs using biocompatible and thermo-sensitive poly(ε-caprolactone)-poly(*N*-vinylcaprolactam-co-*N*-vinylpyrrolidone) [PCL-g-P(NVCL-co-NVP)] copolymers. To the best of our knowledge, no studies exist concerning the study of Dorzolamide- or IMC-loaded PMs based on this type of smart copolymer. These copolymers were previously synthesised by a combination of metal-free ring-opening copolymerisation of the backbone and free radical-controlled RAFT-Madix polymerisation of the grafts [24]. The micellar systems obtained by dialysis in the present study were analysed from both physicochemical and biological points of view.

## 2. Results

### 2.1. Micellar Sizes and Stability

The size of drug-loaded carriers is an important feature that characterises the efficiency of drug delivery systems [25,26]. Compared to low molecular weight surfactant micelles, polymeric micelles (PMs) are much more thermodynamically and kinetically stable. In our previous study [24], it was demonstrated that PCL-g-P(NVCL-co-NVP) copolymers self-assemble in water as spherical micelles with a hydrophobic core formed by the PCL backbone and a hydrophilic corona based on PNVCL and PNVP grafts. In the present study, it was of interest to investigate, first of all, the micellisation of drug-free PMs in buffer solution (PBS, pH = 7.4) and then the influence of drug encapsulation on the micellar size. Table 1 shows the colloidal characteristic values (Z-average, diameter in volume, and the polydispersity index) for free and drug-loaded PMs.

In Table 1, it appears that the sizes of free PMs obtained in PBS (pH = 7.4) are dependent, as expected, on the molecular characteristics of the copolymer samples. The micelles obtained from Cop A, having the lowest total DPn value, have sizes of around 40 nm, whereas Cop B, with higher total DPn, leads to the formation of micelles with sizes around 47 nm. The same tendency was also noted for micelles prepared in water [24]. Moreover, the Dv values follow a similar tendency. The relatively low PDI values are proof that these micelles have a spherical shape and a monodispersed size distribution. 

For the drug-loaded PMs, the micellar sizes have two completely different behaviours as a function of the encapsulated drug. In the case of Dorzolamide, it seems that both the Z-average and Dv values increase at a weight ratio of copolymer/drug of 10:1. This size increase is also accompanied by an increase in PDI values. On the contrary, the encapsulation of IMC led to visible decreases in micellar sizes, whereas the PDI values are almost constant. Nevertheless, both types of drug-loaded PMs have a monomodal distribution, as illustrated in Figure 1. From this figure, it is also possible to notice the micellar size difference between Dorzolamide- and IMC-loaded PMs. If the increase in the micellar size can be easily explained by the fact that the volume of the micellar core increases owing to the presence of drug molecules, the decrease in size can be explained by two hypotheses: the micellar core becomes more compact due to certain interactions between the copolymer and IMC or due to the decrease in a aggregation number, which might be influenced by the more hydrophobic nature of IMC molecules as compared to those of Dorzolamide. 

Concerning the ZP values, also provided in Table 1, the PMs are, within the experimental error limits, almost neutral. This behaviour was expected since both PNVCL and PVP sequences are non-ionic polymers. Nevertheless, a slight decrease in ZP values can be noticed for IMC-loaded PMs, which suggests that IMC molecules might be adsorbed within the micellar corona. 

Table 1 also gives the DEE and DLE values for both drugs. A clear difference appears between the two copolymer samples. Higher DEE and DLE values were calculated for Cop B than for Cop A. Moreover, both the loading and encapsulation efficiencies are higher for IMC than for Dorzolamide. DEE values higher than 70% were determined for the encapsulation of IMC in other types of nanoparticles [27,28]. 

In view of biomedical applications, it is important to determine the stability of the micelles during storage as a function of time. Figure 2 shows the evolution of both the Z-average (Figure 2a) and PDI (Figure 2b) of the drug-free and drug-loaded PMs during storage of the micellar solutions at 4 °C for 30 days

The evolution of the Z-average values given in Figure 2 demonstrates that all of the micellar solutions were stable for at least 30 days of storage at 4 °C. In thermodynamic equilibrium, the system is stable when the free energy has the lowest value. In the case of micellar systems, three factors determine the free energy of the system: (i) the stretching free energy of the micellar core; (ii) the free energy of the interface between the core and the corona of the micelles; and (iii) the free energy of the sequences forming the corona. The almost constant sizes of the PMs are conditioned by the equilibrium between all three free energy contributions [29].

### 2.2. FTIR Spectroscopy

Fourier transform infrared (FTIR) spectroscopy contributes to the identification of the molecular structures of different systems. Changes in IR spectra, such as shifts in absorption bands, band broadening, and the appearance of new absorption bands, will be visible for systems wherein some interactions occur between the polymeric matrix and the encapsulated drug. In the present study, FTIR analysis was carried out in order to investigate the structure of the prepared micellar drug delivery systems.

In the spectrum of Cop B, the characteristic peaks of all three sequences of the copolymers can be identified: CH_2_ stretching, C-N, C-O-C, C-C, CH_2_, and N-C=O appear at 2882.71, 1464.24, 1103.90, 960.9, 840.79, and 526.21 cm^−1^, respectively [30,31].

Further analysing the FTIR spectra in Figure 3, there are obviously differences in the appearance of new bands in the case of micelles loaded with IMC, whereas no differences can be observed for Dorzolamide-loaded PMs, which might be explained by the quite low values of encapsulation efficiency.

Concerning the IMC-loaded PMs, the essential difference is the presence of the 1612 cm^−1^ band due to the C=O groups present in the IMC structure. The presence of the two nuclei explains the prominent band that appears at this wavelength and which is also due to C=C stretching in the aromatic cycle. A new absorption band also occurs at 714.95 cm^−1^, which is attributed to the -C-Cl bond in halogenated compounds, being prominent in aromatic compounds. There is a C-O stretching vibration absorption band at 1195 cm^−1^ due, obviously, to the presence of the hydroxyl group bound to the aromatic nucleus of IMC. Moreover, the characteristic peaks of the PCL-g-P(NVCL-co-NVP) copolymer (at 2882.71, 1103.90, and 960.90 cm^−1^) are shifted, which clearly indicates that some hydrophobic–hydrophobic interactions occur between the polymeric matrix and the IMC during encapsulation.

### 2.3. Assessment of the Haemolysis Degree

Haemolysis is the destruction of red blood cells, along with the release of haemoglobin and other internal components into the surrounding fluid. If this destruction occurs in a significant number of red blood cells in the body, it can lead to dangerous pathological conditions [32,33]. Therefore, all biomedical products designed for intravenous administration should be evaluated for their haemolytic potential [34]. Because the obtained PMs can be used as drug delivery systems, preliminary tests were needed in order to evaluate their interaction with human blood components. The results of the haemolytic toxicity assay of free and drug-loaded PMs after 1.5 and 3 h of incubation at different concentrations are reported in Figure 4.

The results in Figure 4 show that all tested PMs have a haemolysis degree lower than 3% for all tested concentrations at both times. It is important to note that a sample is considered to be haemolytic if the haemolytic percentage is above 5% [35]. Taking into account the obtained results, it can be stated that the tested PMs are suitable for systemic administration.

### 2.4. In Vitro Cytotoxicity Analysis

The MTT assay was performed on fibroblast cells that were in direct contact with three concentrations of PMs in order to evaluate their cytotoxic effects. The in vitro cell viability as a function of time and concentration is provided in Figure 5.

The results of the cytotoxicity assay, given in Figure 5, indicate that the tested PMs at all three established concentrations (10, 50, and 100 µg/mL) did not show cytotoxic effects on fibroblast cells at either 24 or 48 h after incubation. The percentages of cell viability were high, over 90%, in cells that were exposed to a concentration of 10 µg/mL for 24 h for all tested materials. Even at 48 h after incubation, the percentage of cell viability was as high as 90.4% in cells exposed to a concentration of 10 µg/mL for the Cop B sample. For both drug-loaded PMs, the cell viability decreased slightly at all tested concentrations. It also appears that the Dorzolamide-loaded PMs have slightly lower cell viability as compared to IMC-loaded PMs. Micrographs of the fibroblast cells after incubation times of 24 and 48 h are provided in Table 2.

As the cell viability values are all over 80%, the results indicate that the PMs, at the established concentrations, can be used successfully for in vivo biomedical applications.

### 2.5. Drug Release Kinetics

The objective of this study was to obtain an innovative system for the controlled release of two drugs currently used in the treatment of open-angle glaucoma that is capable of delivering controlled and sustained active principles over a longer period of time in order to increase treatment efficiency and patient compliance. The study of drug release kinetics was performed in PBS with pH = 7.4 (close to that of tear secretion) at physiological temperature (37 °C) for 24 h, and the results are displayed in Figure 6.

It was found that free IMC was released quickly, with a pronounced “burst effect” and almost total release (100% efficiency) after about 3 h. Under in vivo conditions, the concentration of the drug that is soluble in an aqueous medium (including the tear fluid) should increase much faster; the delay noted in our case is due to the fact that it must cross the barrier created by the dialysis membrane to accumulate in the physiological fluid outside of it, in which its concentration is determined. However, compared to the micellar systems in which it is encapsulated, the increase in IMC concentration is much faster. It can therefore be appreciated that the duration of action of free IMC under real conditions (in vivo) does not provide a prolonged effect and would require the administration of a new dose of the drug. Free Dorzolamide reaches the maximum efficiency of passage through the membrane (approx. 90%) after 4 h of release due, of course, to its lower solubility in the aqueous medium. Although, kinetically, the process is slower than in the case of IMC, in in vivo test conditions, a new administration would be required after 4–6 h to ensure a therapeutic concentration.

The typical behaviour of controlled-release systems is manifested by micellar systems loaded with drugs: the kinetic curves highlight a pronounced “burst effect” and evolve according to the exponential law characteristic of these systems. The release efficiency of IMC-loaded PMs increases between 0 and 24 h after the same order of encapsulation efficiency. The drug release kinetics is faster for the sample with a copolymer/IMC ratio of 10/1, while the system characterised by the highest copolymer/IMC ratio apparently releases less efficiently (the release efficiency is about 30% after 24 h), but the slope of the curve suggests that the process may continue long after the period for which the kinetic study was performed. In terms of the amount of drug released/g system, the results (based on which the release efficiency was calculated) are close, and it can be stated that they can ensure the desired myotic effect at least over this period of time.

Dorzolamide, whose encapsulation efficiency was lower than that recorded for IMC with a copolymer/drug ratio of 10/1, was released more slowly and in smaller amounts, but release continued slowly, even after 24 h. The lower release efficiency is also the consequence of the lower solubility of the drug in the environment in which the release took place, but given the higher efficacy of treatment with this biologically active principle, which requires the administration of lower amounts of the drug compared to IMC, the system may be considered to be efficient for a duration even longer than 24 h.

All of the results obtained up to this stage confirm the correctness of our initial hypotheses and confirm the possibility of obtaining micellar systems that are loaded with drugs specific to the treatment of open-angle glaucoma and are more effective in treating this condition than the drug administered freely.

### 2.6. In Vivo Evaluation of the IOP

In order to validate the in vitro results, in vivo tests were performed on rabbits to highlight the superiority of our micellar system compared to the free administration of the drugs, as detailed in the following.

IOP measurement, exemplified in Figure 7 for a rabbit from the LE4 group, was performed in all animals at different times of the day, knowing that IOP varies at different times of the day (morning, noon, and evening), both diurnal and nocturnal. Six measurements were obtained for each eye and for each moment of the day, and the average IOP value was calculated by the device and reported in millimetres of mercury (Table 3).

In vivo results showed that the LE3 and LE4 groups experienced a decrease in IOP from 9–12 mmHg to 7–9 mmHg (in the case of LE3) and from 8–12 mmHg to 4–5 mmHg (for LE4), demonstrating that the drug-loaded micellar system has a sustained IOP-lowering action. Even at the end of the experimental period, the decrease in IOP was maintained, which means that the PMs continued to release the drug for a longer period of time than the conventional topical treatment. This effect is probably due to two aspects of this formulation. First of all, the mucoadhesive properties of the micelles in which the drug was incorporated determined their longer contact time on the surface of the cornea, and due to their small size, some of them penetrated the cornea (effect demonstrated by histological analysis, not presented here). Second, the PMs altered the rate and extent of Dorzolamide and Indomethacin release and absorption, resulting in prolonged IOP reduction. The results obtained are in agreement with those reported by Prabhu et al. [36], who developed an improved formula in the form of drug-carrying niosomes, which led to a decrease in IOP for a longer period of time. The better performance of drugs encapsulated in PMs is manifested by their higher bioavailability. This increased bioavailability of Dorzolamide from the PMs is the overall result of the longer contact time, the controlled release of dorzolamide from the micellar complex, and the ability of the micelles to act as a penetration enhancer due to their chemical content.

## 3. Materials and Methods

### 3.1. Materials

Hydrophobic Dorzolamide and IMC were purchased from Sigma Aldrich (St. Louis, MO, USA). All solvents were used as received without further purification. The human dermal fibroblast cell line (HDFa) and the necessary supplies (antibiotic cocktail: penicillin and streptomycin; non-essential amino acids; trypsin solution; and foetal bovine serum—FBS) for the in vitro cytotoxicity assay were purchased from Thermo Fisher Scientific (Waltham, MA, USA).

Two copolymer samples were investigated in the present study. The PCL-g-P(NVCL-co-NVP) copolymers were obtained by a grafting-from technique in three steps, as previously reported [24]. The backbone was synthesised by ring-opening copolymerisation of ε-caprolactone (ε-CL) and α-chloro-ε-caprolactone (α-Cl-ε-CL). In the second step, a RAFT macroinitiator was prepared by the substitution of pendant chloro groups with a xanthate salt. Finally, RAFT-Madix copolymerisation of N-vinyl caprolactam (NVCL) with N-vinylpyrrolidone (NVP) was carried out. The obtained copolymers were analysed by ^1^H NMR (Bruker AC-400F, MA, USA) and size exclusion chromatography (Shimadzu LC-20AD liquid chromatograph, Japan), and the molecular characteristics of the two samples are provided in Table 4.

### 3.2. Methods

#### 3.2.1. Micelle Preparation Procedure

The preparation of PMs was carried out by a dialysis method starting from a common solvent. In a typical procedure, 100 mg of copolymer was added to 10 mL of dimethylsulfoxide (DMSO) solution and stirred at room temperature until complete dissolution of the copolymer. Afterwards, the solution was dialysed against 1 L of ultrapure water using cellulose dialysis membranes (molecular weight cut-off: 12 kDa; manufacturer: Sigma Aldrich, Steinheim, Germany). The water was changed eight times during 24 h of dialysis. The dry powder was collected after the freeze-drying of the micellar solutions and then was stored at −4 °C before further use.

A similar procedure was used for the preparation of Dorzolamide- and IMC-loaded PMs, with the difference that the solution of the block copolymer in DMSO was added to different amounts of drugs in order to have three weight ratios between the copolymer and the drug: 10:1; 5:1, and 2:1.

#### 3.2.2. Physicochemical Characterisation Methods

The micellar sizes were investigated by dynamic light scattering (DLS) measurements in phosphate buffer solution (PBS; pH = 7.4), as this medium is similar to the in vivo medium. DLS was carried out on a Malvern Zetasizer Pro (Malvern, Worcestershire, UK) using NIBS (Non-Invasive BackScattering) technology, equipped with a 4mW He–Ne laser operating at a wavelength of 532 nm and at a scattering angle of 173°. The software package of the instrument calculates, by using the Stokes–Einstein equation, the hydrodynamic diameter (volume average) Dv, the Z-average diameter, which is an intensity-weighted size average, and the polydispersity index (PDI) of the sample. In order to determine the mean diameter of the particles, the data were collected in automatic mode, typically requiring a measurement duration of 70 s. For each experiment, 5 consecutive measurements were carried out. The stability of the micellar solutions stored at 4 °C was assessed as a function of time for 30 days. The zeta potential values of PMs were determined by electrophoresis in phosphate buffer solution (PBS; pH = 7.4) using the same instrument. 

In order to determine the encapsulation efficiency of both drugs, two calibration curves were constructed in DMSO using different concentrations of Dorzolamide and IMC, and their absorbance values were recorded on a Nanodrop spectrophotometer (Nanodrop One, Thermo Scientific, Waltham, MA, USA) at a wavelength of 260 nm for Dorzolamide and 270 and 320 nm for IMC. A known amount of the drug-loaded micelles in powder form was dissolved in 1 mL of DMSO in order to completely destroy the micelles and to release the loaded drug. The amount of both Dorzolamide and IMC from the micelles was spectrophotometrically quantified based on their calibration curves using a UV spectrometer (Nanodrop One, Thermo Scientific, Waltham, MA, USA). Drug encapsulation efficiency (DEE) and drug loading efficiency (DLE) were calculated using Equations (1) and (2), respectively:(1)$$\mathrm{DEE}\;(\%)=\frac{\mathrm{amount}\;\mathrm{of}\;\mathrm{drug}\;\mathrm{in}\;\mathrm{micelles}}{\mathrm{amount}\;\mathrm{of}\;\mathrm{added}\;\mathrm{drug}}\times 100.\;$$
(2)$$\mathrm{DLE}\;(\%)=\;\frac{\mathrm{amount}\;\mathrm{of}\;\mathrm{drug}\;\mathrm{in}\;\mathrm{micelles}}{\mathrm{amount}\;\mathrm{of}\;\mathrm{added}\;\mathrm{polymer}\;\mathrm{and}\;\mathrm{drug}}\;\times \;100$$


Three determinations were performed for each sample, and the errors were ±0.3%.

Fourier transform infrared spectroscopy (FTIR) vibrational spectra of lyophilised Dorzolamide- and Indomethacin-loaded PMs and free PMs were recorded using an attenuated total reflectance (ATR) device with a Shimadzu spectrometer (IRSpirit, Kyoto, Japon) in the scanning range of wavelength 400–4000 cm^−1^. The number of scans and the resolution were fixed to 50 and 4 cm^−1^, respectively.

In vitro drug release kinetics was studied in phosphate buffer solution (PBS) at 37°C and pH = 7.4, a value that is similar to that of blood. For that, a given amount of lyophilised drug-loaded micellar powder (10 mg) was dispersed in 5 mL of PBS, and then the suspension was added to a dialysis membrane. The samples thus prepared were immersed in 20 mL of PBS with the corresponding pH value under stirring at 37 °C. At defined time intervals, samples were extracted in order to spectrophotometrically quantify the amounts of released drugs using the Nanodrop spectrophotometer (Nanodrop One, Thermo Scientific, Waltham, MA, USA). The drug release efficiency was calculated using Equation (3):(3)$$R_{ef}\left(\%\right)=\frac{m_{r}}{m_{l}}\times 100$$
where *m_r_* is the amount of drug released from PMs (mg); *m_l_* is the amount of drug encapsulated into PMs (mg).

#### 3.2.3. In Vitro Biological Characterisation Methods

The haemolytic potential of the obtained PMs was evaluated using a spectrophotometric method adapted from Rata et al. [37]. These tests were started after obtaining the institutional ethical authorisation and the appropriate informed consent. Blood from healthy non-smoking human volunteers was collected in vacutainer tubes and treated with PMs. In total, 5 mL of anti-coagulated blood was centrifuged at 2000 rpm (RCF = 381× *g*) for 5 min and washed with normal saline solution several times to completely remove the plasma and obtain erythrocytes. After purification, erythrocytes were re-suspended in 25 mL of normal saline solution. PM saline solution with different concentrations (0.5 mL) was added to 0.5 mL of erythrocytes suspension (final concentrations were 10, 50, 100, and 200 mg PMs/mL erythrocyte suspension). Positive (100% lysis) and negative (0% lysis) control samples were prepared by adding equal volumes (0.5 mL) of Triton X-100 and a standard saline solution. The samples were incubated at 37 °C for 180 min. Once every 30 min, the samples were gently shaken to re-suspend erythrocytes and PMs. After the incubation time, the samples were centrifuged at 2000 rpm (RCF = 381× *g*) for 5 min, and 100 µL of supernatant was incubated for 30 min at room temperature to allow haemoglobin oxidation. Oxyhaemoglobin absorbance in supernatants was measured at 540 nm using a Nanodrop One UV-Vis Spectrophotometer (Thermo Fischer Scientific, Waltham, MA, USA). All samples were analysed in triplicate. The haemolytic percentage was calculated using Equation (4):(4)$$\mathrm{Haemolysis}\;(\%)=\;\frac{\left(A_{S}-A_{NC}\right)}{\left(A_{PC}-A_{NC}\right)}\times 100$$
where *A_S_* is the absorbance of the sample; *A_NC_* and *A_PC_* are the absorbance values of the negative and positive controls, respectively.

The MTT method was applied to assess the in vitro cytotoxicity of free and drug-loaded PMs by using adherent adult human fibroblast cells of dermal origin (HDFa). After thawing the fibroblast cells in the thermostatic bath at 37 °C (digital thermostatic baths, DIGIBATH 2 − BAD\2RAYPA, Spain), the cells (HDFa) were cultured in complete growth medium: DMEM (Dulbecco’s Modified Eagle Medium) supplemented with 10% foetal bovine serum, 1% antibiotics, and 1% non-essential amino acids at 37 °C in a humidified atmosphere of 5% CO_2_ (MCO-5AC CO_2_ Incubator, Panasonic Healthcare Co., Ltd., Sakata Oizumi-Machi Ora-Gun Gunma, Japan). Cells were allowed to proliferate in culture flasks (NuncTM EasYFlask 25 cm^2^ TM, ThermoFisher Scientific, Roskilde, Denmark) to reach 80% confluence and then were trypsinised with 0.05% trypsin solution at 37 °C, followed by the addition of complete medium to neutralise the trypsin; cells were centrifuged (Rotofix-32A, Hettich, Andreas GmbH Hettich & Co.KG, Tuttlingen, Germany) and re-suspended in complete DMEM medium. For performing the in vitro cytotoxicity assay, the reagents were purchased from Thermo Fisher Scientific. After centrifugation and re-suspension in fresh medium, viable cells were plated in flat-bottom 96-well plates (TPP Techno Plastic Products AG, Trasadingen, Switzerland) and incubated for 24 h. After 24 h of incubation, the culture growth medium was replaced with fresh medium, and the test materials were put in direct contact with the fibroblast cells at 3 concentrations: 10, 50, and 100 µg/mL. Prior to performing the cytotoxicity test, the materials were sterilised with UV-VIS radiation for 3 min. Microscopic analysis was performed using an inverted optical microscope (CKX41, Olympus, Tokyo, Japan) with a built-in camera and QuickPHOTO camera 3.0 software (Olympus, Tokyo, Japan). Cell viability was determined by the quantitative colorimetric assay 24 and 48 h after incubation of cells treated with tetrazolium salt (3-(4,5-dimethylthiazol-2-yl)-2,5-diphenyltetrazolium bromide (MTT)) (Merck Millipore, Darmstadt, Germany). After 24 h incubation, 100 μL of culture medium was replaced with 100 μL of fresh medium, followed by the addition of 10 μL of MTT dye to each well and incubation for 4 h at 37 °C in 5% CO_2_. After 4 h of incubation, 90 μL of medium was removed from each well, and 100 μL of DMSO was added to dissolve the formazan crystals, followed by re-incubation for 10 min at 37 °C. The absorbance was measured at 570 nm using a Multiskan FC automatic plate reader (Thermo Fisher Scientific, Oy, Finland) with Sknalt Software 4.1 (Fisher Scientific, Oy, Finland). Each sample was tested in triplicate, and cell viability was expressed as % of untreated cells (control), considered 100% viable.

#### 3.2.4. In Vivo Tests

Rabbits were divided into 6 groups (LC—control group; LE1–LE5, experimental groups), with each group being formed by 3 rabbits (6 tested eyes). All experimental groups (LE1–LE5) received topical treatment with dexamethasone 4 times/day for a period of 15 days to induce an increase in intraocular pressure (IOP). The rabbits in the LE5 group were also injected subconjunctivally with dexamethasone to induce an increase in IOP in a shorter period of time.

During the 15 days, the eyes were periodically inspected for signs of inflammation, and IOP was measured using the iCare 100 non-contact tonometer (iCare Finland). Biomicroscopically, slight corneal oedema and, in some cases, mild conjunctival congestion were also detected. LC (the control group) did not receive any drug treatment.

After 15 days, treatment with the drug-loaded micelles was carried out for another 15 days, after which the animals were sacrificed and the eyes were enucleated for histological analysis, which is not reported in this work.

LE1—received topical treatment 2 times/day with AL1 (empty PMs).

LE2—received topical treatment 2 times/day with AL2 (IND-loaded PMs).

LE3—received topical treatment 2 times/day with AL3 (Dorzolamide-loaded PMs).

LE4—received topical treatment 2 times/day with AL4 (IND- and Dorzolamide-loaded PMs).

LE5—received topical and subconjunctival dexamethasone treatment.

## 4. Conclusions

This paper reports results on the preparation and use of a smart drug delivery system based on the use of thermo-sensitive amphiphilic poly(ε-caprolactone)-poly(*N*-vinylcaprolactam-co-*N*-vinylpyrrolidone) [PCL-g-P(NVCL-co-NVP)] graft copolymers capable of forming micelles, in which two specific drugs for the treatment of open-angle glaucoma were loaded, namely, Indomethacin (IMC) and Dorzolamide. The free and drug-loaded PMs were characterised by DLS in terms of their size, stability, and zeta potential. It was found that Dorzolamide increased the micellar diameter, whereas IMC had the opposite effect. The drug-loaded PMs showed high stability over time, proven by the slight evolution of their sizes and PDI values over 30 days. FTIR spectroscopy demonstrated the ability of micelles to encapsulate drugs by the presence of their specific absorption peaks in the spectra of drug-loaded PMs and also some interactions between the polymeric matrix and the loaded IMC. However, no interactions were visible in the case of Dorzolamide. In vitro tests proved both their non-cytotoxic behaviour, with cellular viabilities higher than 80%, and low degree of haemolysis (less than 3% even at high concentrations such as 200 μg/mL), as well as the ability of this new system to release the encapsulated drugs over an extended period of time. In vivo tests on rabbit eyes validated the results obtained in vitro, proving the ability of the designed micellar system to release the drugs in a sustained manner, producing a sharp and long-lasting decrease in intraocular pressure. Based on the obtained results, it can be inferred that these PMs might be recommended as safe systems for further in vivo tests.

## Figures and Tables

**Figure 1 ijms-23-09382-f001:**
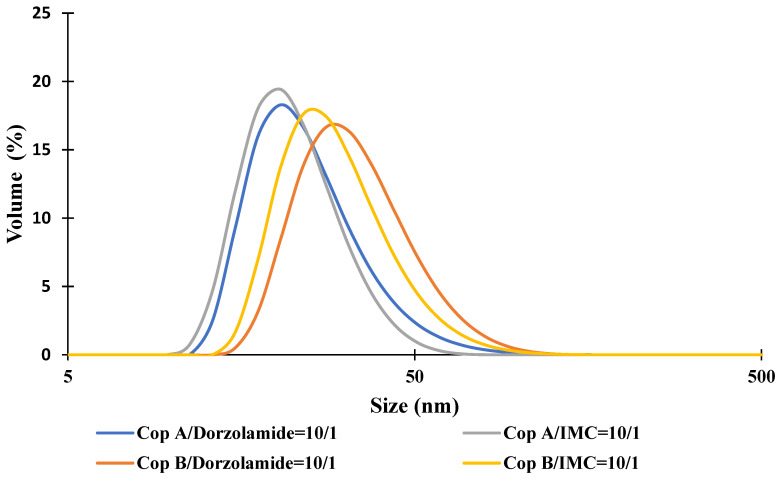
Size distribution curves for drug-loaded PMs at a copolymer/drug ratio of 10/1.

**Figure 2 ijms-23-09382-f002:**
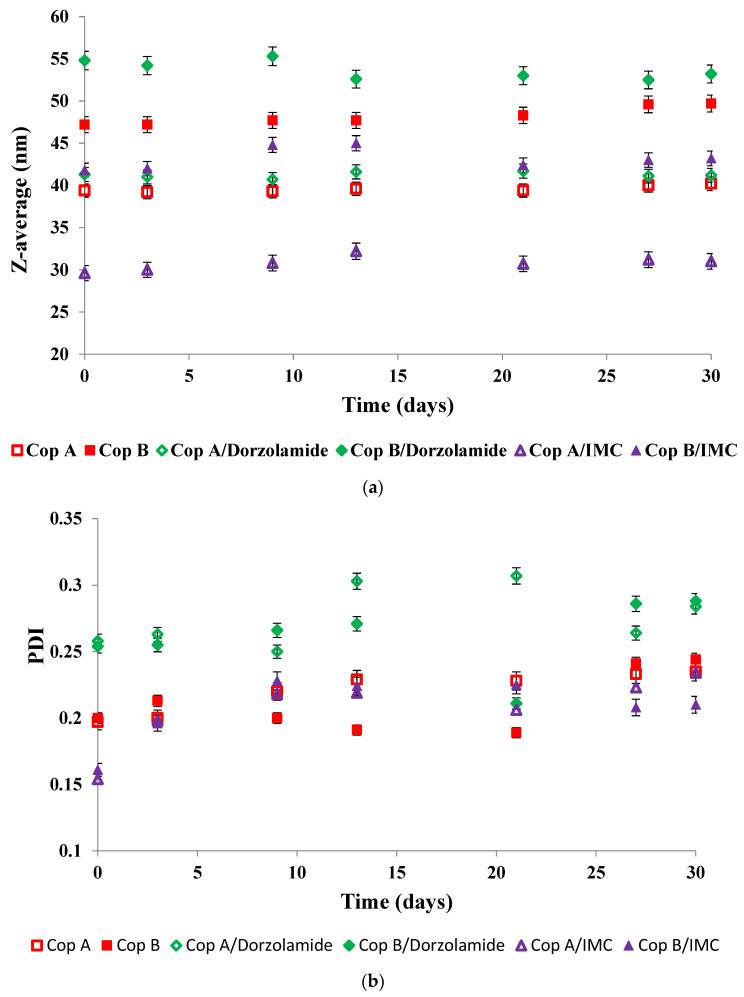
Evolution of both Z-average and PDI as a function of time for drug-free (**a**) and drug-loaded (**b**) PMs at a copolymer/drug ratio of 10/1.

**Figure 3 ijms-23-09382-f003:**
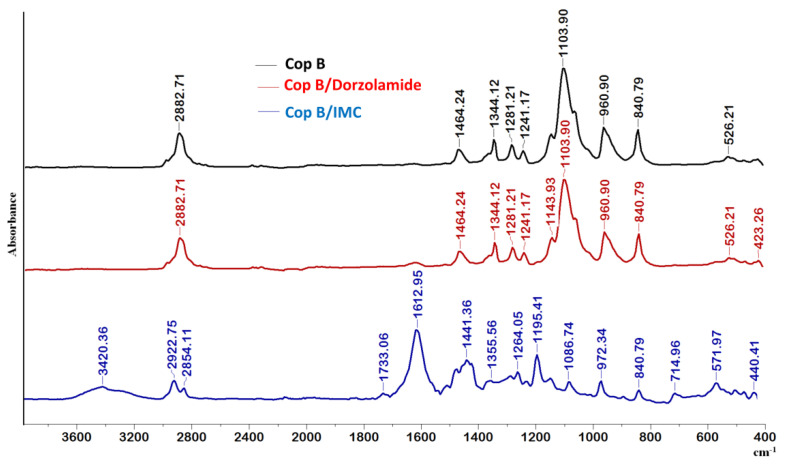
FTIR spectra of free and drug-loaded PMs at a copolymer/drug ratio of 10/1.

**Figure 4 ijms-23-09382-f004:**
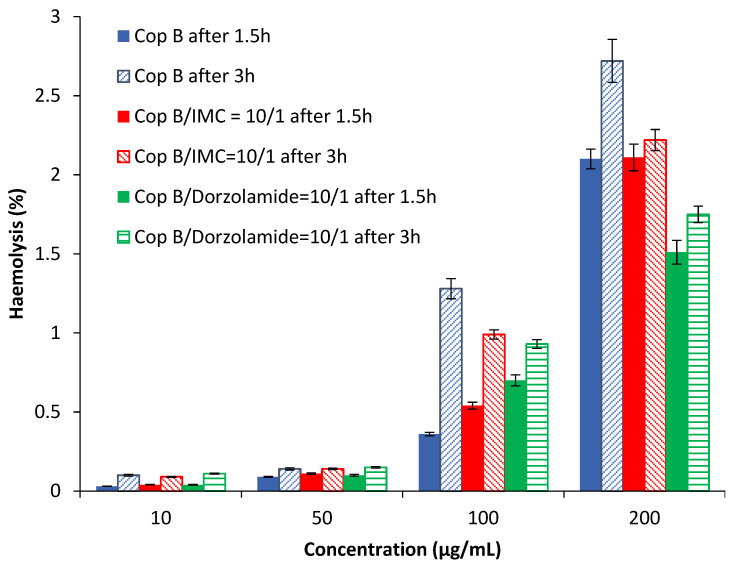
Haemolysis degree as a function of the PM concentration.

**Figure 5 ijms-23-09382-f005:**
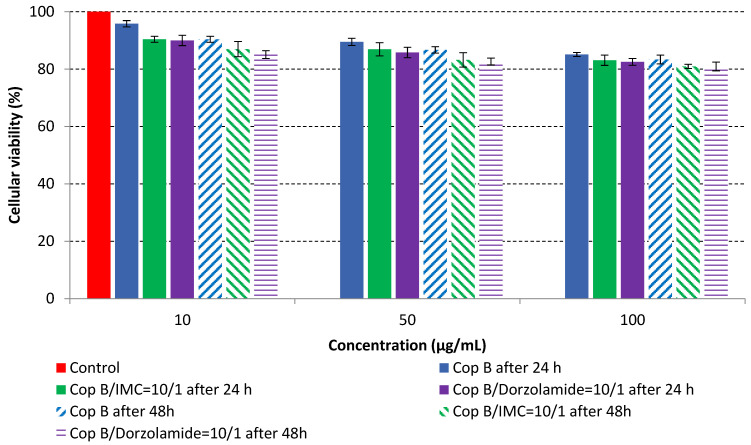
In vitro cell viability of PMs as a function of concentration and time.

**Figure 6 ijms-23-09382-f006:**
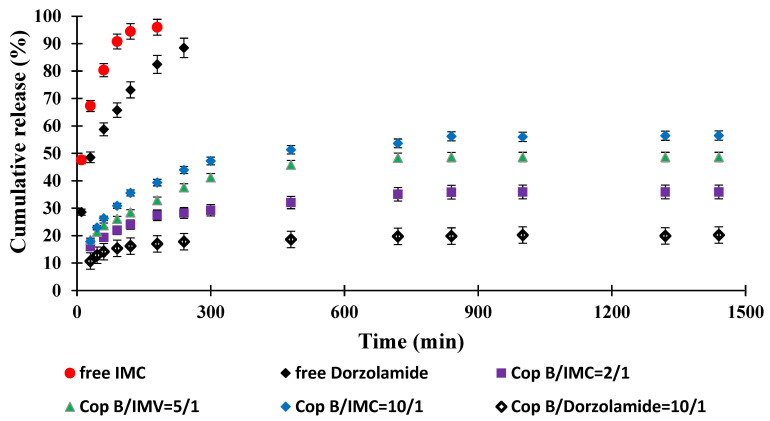
Drug release kinetics in PBS (pH = 7.4) at 37 °C.

**Figure 7 ijms-23-09382-f007:**
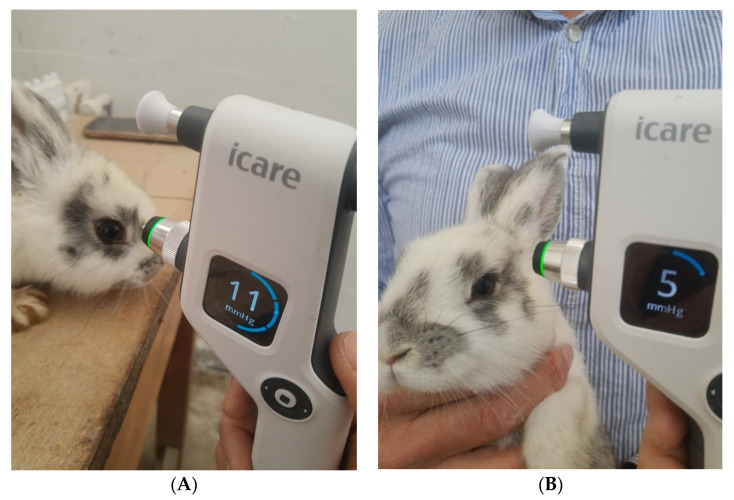
IOP measurement with the iCare non-contact tonometer for a rabbit from the LE4 group and its values after the administration of dexamethasone (15 days, (**A**)) and after the administration of drugs encapsulated in micelles (15 days, (**B**)).

**Table 1 ijms-23-09382-t001:** Colloidal characteristics of freshly prepared suspensions of Cop A and Cop B micelles in PBS (pH = 7.4) at 25 °C, at a concentration of 0.5 wt%, in the absence of drug, and with a copolymer/drug ratio of 10/1.

Sample	Z-Average (nm)	Dv (nm)	PDI	ZP (mV)	DEE (%)	DLE (%)
**Cop A**	39.4 ± 0.1	24.2 ± 0.2	0.197	−3.5	-	-
**Cop A/Dorzolamide = 10/1 (wt/wt)**	41.3 ± 0.4	24.5 ± 0.5	0.258	−3.3	20.2	4.4
**Cop A/IMC = 10/1 (wt/wt)**	29.6 ± 0.3	23.0 ± 0.1	0.154	−7.5	68.5	6.3
**Cop B**	47.2 ± 0.2	34.8 ± 0.4	0.200	−3.3	-	-
**Cop B/Dorzolamide = 10/1 (wt/wt)**	54.8 ± 0.3	37.5 ± 0.2	0.354	−3.3	34.0	6.7
**Cop B/IMC = 10/1 (wt/wt)**	41.8 ± 0.4	31.0 ± 0.1	0.161	−5.5	75.1	8.2
**Cop B/Dorzolamide = 10/1+Cop B/IMC = 10/1** **(50/50 wt/wt)**	45.7 ± 0.5	33.9 ± 0.2	0.252	−4.1	-	-

**Table 2 ijms-23-09382-t002:** Micrographs of fibroblast cells after 24 and 48 h incubation times, respectively.

Sample		24 h	48 h
Control		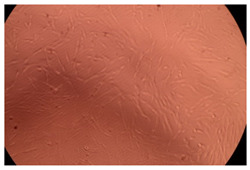	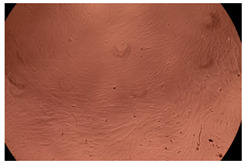
Cop B	10 µg/mL	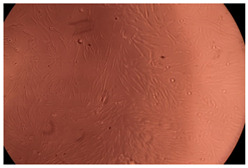	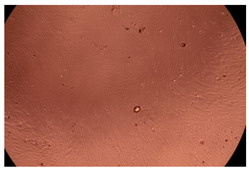
	50 µg/mL	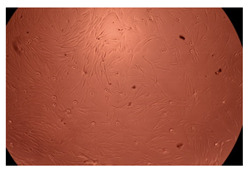	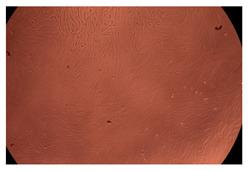
	100 µg/mL	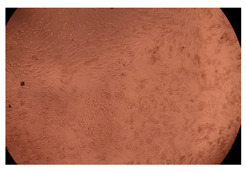	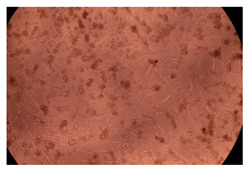
Cop B/IMC = 10/1 (wt/wt)	10 µg/mL	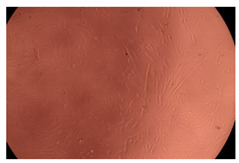	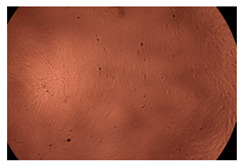
	50 µg/mL	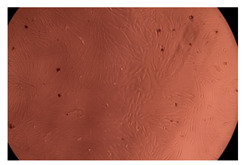	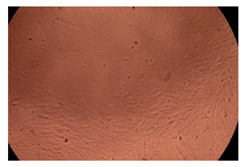
	100 µg/mL	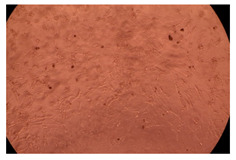	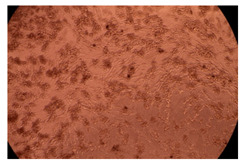
Cop B/Dorzolamide = 10/1 (wt/wt)	10 µg/mL	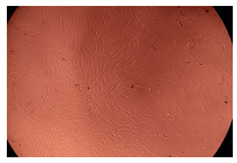	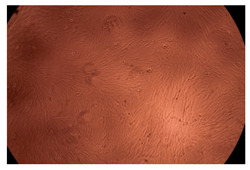
	50 µg/mL	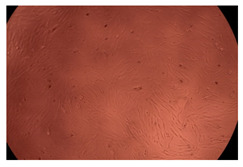	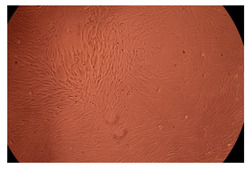
	100 µg/mL	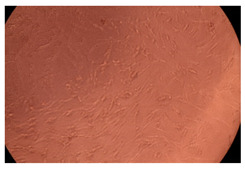	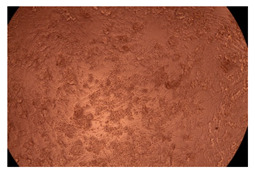

**Table 3 ijms-23-09382-t003:** Evolution of IOP values over 30 days.

Sample	IOP (mmHg)					
	After 15 Days of DEX Administration			After 15 Days of Treatment		
	Morning	Noon	Evening	Morning	Noon	Evening
LC	5	6	7	5	6	7
LE1	9	10	11	9	10	11
LE2	9	10	11	9	10	11
LE3	9	10	11	7	8	9
LE4	9	10	11	4	5	5
LE5	9	11	12	10	11	13

**Table 4 ijms-23-09382-t004:** Molecular characteristics of the PCL-g-P(NVCL-co-NVP) copolymers.

Copolymer	Sample Name	PNVCL (mol %)	PNVP (mol%)	DPn, Average per Graft
PCL_120_-g-P(NVCL_507_-co-NVP_128_)	Cop A	64	20	42
PCL_120_-g-P(NVCL_1253_-co-NVP_139_)	Cop B	82	10	93

## Data Availability

The data presented in this study are available on request from the corresponding author.

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
