# Peer review of "Drug-Loaded Polymeric Micelles Based on Smart Biocompatible Graft Copolymers with Potential Applications for the Treatment of Glaucoma"

_ijms, 2022, doi:10.3390/ijms23169382_

Round 1
Reviewer 1 Report
The work by Ozturk et al reports drug-loaded polymeric micelles as potential glaucoma therapeutics. The work is of interest to the field and may be accepted after addressing the points below:
- Figures for the data shown in Table 2 should be presented in the supplementary information file.
-There are no error bars in all data points of Figures 2, 4, and 6 in the manuscript.
Author Response
Reviewer 1
The work by Ozturk et al reports drug-loaded polymeric micelles as potential glaucoma therapeutics. The work is of interest to the field and may be accepted after addressing the points below:
- Figures for the data shown in Table 2 should be presented in the supplementary information file.
Even if the data presented in fig 2 is not essential for this study, we prefer to keep this figure in the main text.
-There are no error bars in all data points of Figures 2, 4, and 6 in the manuscript.
As suggested by the reviewer, we added the error bars.
Reviewer 2 Report
Just one comment. Couldn't find the name / company for apparatus used for drug release studies. However it is a minor thing and I am sure Authors could add it during the proofreading. It is a go. Congrats!
Author Response
Reviewer 2
Just one comment. Couldn't find the name / company for apparatus used for drug release studies. However it is a minor thing and I am sure Authors could add it during the proofreading. It is a go. Congrats!
We would like to thank the reviewer for his congratulations. Drug release studies were carried out by dialysis and the amount of released drugs was determined using the Nanodrop spectrophotometer (Nanodrop One, Thermo Scientific, Waltham, MA, USA).
This manuscript is a resubmission of an earlier submission. The following is a list of the peer review reports and author responses from that submission.
Round 1
Reviewer 1 Report
The present article in the title presents an application for the developed nanosystem but there is no test in the manuscript in which it has been evaluated either in vitro/ex vivo or in vivo. The in vitro assays presented are not relevant for topical ocular application. Thus, many results are still lacking for this work to be considered for publication. There are many gaps from the evaluation part of the formulation either at the development level or in the evaluation of effectiveness or even in the proof of concept. Therefore, I consider that in the present form this work is not in a condition to be published since it does not add innovation or novelty.
Reviewer 2 Report
I wanted to thank Authors for compying with my previous remarks - this is more clear now what kind of dosage form we deal with. To complete the picture I just ask to describe the in vitro studies method in more details: what apparatus (vendor, settings) etc.